# Quantitative Assessment of Fixational Disparity Using a Binocular Eye-Tracking Technique in Children with Strabismus

**DOI:** 10.3390/jemr18020006

**Published:** 2025-03-10

**Authors:** Xiaoyi Hou, Xubo Yang, Bingjie Chen, Yongchuan Liao

**Affiliations:** Department of Ophthalmology, West China Hospital, Sichuan University, Chengdu 610041, China; houx@yahoo.com (X.H.); cocayang@163.com (X.Y.); mellrince@163.com (B.C.)

**Keywords:** eye movement, fixational disparity, strabismus, vision science

## Abstract

Fixational eye movements are important for holding the central visual field on a target for a specific period of time. In this study, we aimed to quantitatively assess fixational disparities using binocular eye tracking in children with strabismus (before and after surgical alignment) and healthy children. Fixational disparities in 117 children (4–18 years; 57 with strabismus and 60 age-similar healthy controls) were recorded under binocular viewing with corrected refractive errors. Disparities in gaze positions relative to the target location were recorded for both eyes. The main outcome measures included fixational disparities along horizontal and vertical axes in the fixation test. Children with strabismus exhibited significant (*p* < 0.001) fixational disparities compared to healthy children in both horizontal and vertical directions. Additionally, children with esotropia had poorer fixational function compared to those with exotropia. The occurrence of fixational disparities significantly decreased in the horizontal direction following strabismus surgery. A significant negative correlation was observed between binocular best-corrected visual acuity and fixational disparities in children with strabismus. Children with strabismus had significant fixational disparities that were observably diminished in the horizontal direction after surgical alignment. Binocular assessment of fixational disparities can provide a more comprehensive evaluation of visual function in individuals with strabismus.

## 1. Introduction

Fixational eye movements refer to a series of spontaneous, controlled, and autonomous eyeball movements that are vital for holding the central visual field on a target for a specific period of time. Fixational disparity is an important measure in the study of fixational eye movements and binocular vision. The term refers to the minor misalignment of the eyes when trying to fixate on a single point. This misalignment is usually too small to cause double vision but can be an indicator of how well the eyes work together to maintain focus [1,2].

Research has shown that fixational disparity is closely related to the dynamics of vergence eye movements, which are responsible for aligning the eyes in response to differences in the position of objects [3,4]. For instance, fixational disparity has been described as a steady-state error of vergence, meaning that it reflects a residual error after the eyes have attempted to correct for the disparity [5,6,7,8,9,10,11].

Strabismus during development can lead to abnormal visual experiences in both eyes, disrupting normal binocular visual function [12,13]. The authors of previous studies have reported abnormal fixational eye movements in patients with strabismus [14,15]. While laboratory eye-tracking equipment for measuring fixational eye movements provides high temporal and spatial resolution in addition to a wide range of metric data, these systems are often prohibitively expensive and complex, rendering them impractical for everyday clinical use [16].

Therefore, in this study, we propose a low-cost and easy-to-use method to assess fixational disparities as an indicator of fixational eye movements using a binocular eye-tracking technique in children with strabismus and healthy children, in addition to use in children with strabismus before and after corrective surgery.

## 2. Materials and Methods

### 2.1. Participants

This study included pediatric participants aged 4 to 18 years who attended the Department of Ophthalmology at West China Hospital of Sichuan University between 1 June 2020, and 31 December 2022. Participants were categorized into the strabismus group or the control group. Guardians provided written informed consent, while children offered verbal or written assent according to their age. The study adhered to the principles of the Declaration of Helsinki and received approval from the Institutional Review Board of West China Hospital.

### 2.2. Design

All participants underwent standardized ophthalmological examinations, including the following:

Best-corrected visual acuity measured using the Logarithm Visual Acuity Chart with high-contrast Optotype; cycloplegic refraction (1% atropine for esotropia; 0.5% tropicamide for exotropia and healthy children); ocular alignment assessed using the prism plus alternate cover test; anterior segment evaluation via slit lamp examination; stereopsis measured using the Titmus stereo test (Stereo Optical Co., Inc. Chicago, IL, USA); fundus examination; and binocular eye movement evaluation using the study’s binocular paradigm. All participants underwent refractive correction, and measurements of fixation disparity and strabismus angles were taken both before and after correction to ensure data accuracy.

In the strabismus group, stereoacuity was assessed both preoperatively and postoperatively. Participants were eligible for inclusion if they had no history of ocular trauma or ocular pathology (e.g., nystagmus, cataract, or ptosis), no systemic diseases (self-reported), no attentional dysfunctions such as attention deficit disorder (ADD) or other neurological conditions (e.g., concussion or learning disabilities), no prior intraocular surgery, and no amblyopia based on the 2022 Amblyopia “PPP” guideline [17]. Additionally, participants needed to demonstrate sufficient cooperation during examinations.

### 2.3. Inclusion Criteria

The strabismus group included participants who met the following criteria: diagnosis of concomitant strabismus, including esotropia or exotropia; postoperative ocular alignment within ±8Δ at 6 m and 33 cm fixation and completion of at least one postoperative follow-up.

The control group included participants who fulfilled the following criterion: normal ocular alignment (≤±5Δ at 6 m and 33 cm fixation) and normal binocular eye movement function.

### 2.4. Exclusion Criteria

Participants were excluded from the study if they had attentional dysfunction (e.g., attention-deficit disorder or attention-deficit/hyperactivity disorder), neurological conditions (e.g., concussion, traumatic brain injury, epilepsy, or learning disabilities), significant ocular pathologies (e.g., cataracts, glaucoma, or retinal disorders), systemic conditions (e.g., Down syndrome, cerebral palsy, or other developmental disorders), or an inability to cooperate with the eye-tracking procedure due to behavioral or cognitive limitations. These exclusion criteria were implemented to ensure that the study results were not confounded by factors unrelated to strabismus or fixational disparity.

### 2.5. Materials

A Tobii Eye Tracker 5 (Tobii, Stockholm, Sweden) was utilized to detect fixational disparities. The experiment was conducted in a quiet, private room with consistent natural illumination. Visual stimuli were displayed on a 32-inch three-dimensional (3D) monitor (LG Electronics, Seoul, Republic of Korea) with a resolution of 1920 × 1080 pixels, a refresh rate of 120 Hz, and a spatial resolution of 0.3 degrees (minimum 0.1 degrees) at a viewing distance of 80 cm. Participants wore 3D polarized glasses with spectacle correction if required and were seated on a non-wheeled, height-adjustable chair, ensuring their eyes were level with the screen center. Gaze positions were recorded at 120 Hz using the remote eye tracker. A trial adaptation period was implemented before formal measurements to acclimate participants to the testing environment and stabilize their visual state.

### 2.6. Inspection Procedure

Fixational disparities were measured under static binocular-viewing conditions. Participants fixated on a bright blue dot (1.4° diameter) with a black cross-shaped center against a black background. The target appeared sequentially at 9 locations: 8 peripheral points on a circle (8.3° radius) and 1 central point. Each target remained visible for 3 s before automatically switching to the next location. Once stable fixation was confirmed, the researcher manually initiated recording.

At each target location, the eye tracker extracted 10 gaze position samples. The mean horizontal and vertical disparities relative to the target location were calculated for each eye and recorded as binocular mean values (Figure 1a,b).

In the strabismus group, fixational disparities were measured preoperatively and again one month postoperatively. The mean horizontal and vertical deviations (in pixels) from both eyes were calculated at each target location. Red numbers around the center represented extracted gaze positions. Significant fixational disparities were observed in the strabismus group preoperatively, while postoperative results were comparable to those of the control group (Figure 1c,d).

### 2.7. Statistical Analysis

Statistical analyses were performed using SPSS (version 26.0, IBM Corp., Armonk, NY, USA). The number of shifted pixels in fixational disparities was treated as n-dimensional data and expressed as the mean (standard deviation). To address potential issues regarding multiple comparisons, appropriate corrections were applied where necessary. The normality of the data distribution was assessed using statistical tests and visual inspections. For data that followed a normal distribution, parametric tests were used; for data that did not follow a normal distribution, non-parametric tests were employed. Specifically, the Mann–Whitney U test was used for comparisons between groups, and paired sample tests (e.g., paired *t*-tests or Wilcoxon signed-rank tests, depending on the distribution) were used to compare internal differences within the same group. Differences were considered statistically significant at *p* < 0.05.

## 3. Results

A total of 117 participants were recruited for this study. The control group comprised 60 participants with an average age of 7.37 [0.20] years, including 26 males (43.3%) and 34 females (56.7%). The strabismus group consisted of 57 individuals with an average age of 7.76 [3.59] years, including 36 males (62.1%) and 21 females (37.9%). There were no statistically significant differences in age (*p* = 0.82) and sex (*p* = 0.78) between the two groups (Table 1).

The results of the Titmus stereo test showed that the average score of stereoacuity for the strabismus group before surgery was 243.38″ [40.44] compared with the average score of 50.33″ [4.83] for the control group. The score of the strabismus group was significantly higher than that of the control group (*p* = 0.02). In the strabismus group, the preoperative score was higher than the postoperative score (221.29″ [40.19]), and the difference was statistically significant (*p* < 0.001) (Table 2).

The Kruskal–Wallis test was used for comparison among the three groups, in which *p*-values < 0.001 (represented by ***) and *p*-values < 0.01 (represented by **) indicated significant differences in pairwise comparisons and *p*-values > 0.05 indicated no significant differences. The average disparity for both the *X*-axis and *Y*-axis in the strabismus group was significantly greater than that of the control group before surgery (*p* < 0.001). After surgery, the *X*-axis disparity showed a significant difference compared to the control group; in comparison, the *Y*-axis disparity no longer exhibited a significant difference. The error bars in Figure 2 represent the standard error of the mean (SEM), which provides an estimate of the uncertainty around the mean values (Figure 2).

In the strabismus group, the average disparity on the *X*-axis of participants with exotropia was significantly smaller than that of participants with esotropia before surgery (*p* = 0.004), with no significant difference being observed after surgery (*p* = 0.84) (Table 3).

Next, we present a comparison of the main outcome measures across groups. *X*-axis/*Y*-axis: Pixel along the horizontal/vertical axis in the fixational disparity test (error bars include means and standard deviations). In horizontal (*X*-axis) fixation, the average score of participants with exotropia was significantly lower than that of participants with esotropia before surgery, although no significant difference was observed post-surgery. In vertical (*Y*-axis) fixation, no significant differences were found between participants with exotropia and participants with esotropia both during the pre- and post-surgery period (Figure 3).

Pearson correlation analysis was performed between fixational disparities and various clinical characteristics before surgery, including stereoacuity (*x*-axis: Figure 4a, *y*-axis: Figure 4e), best-corrected visual acuity (BCVA; *x*-axis: Figure 4b for right eyes, Figure 4c for left eyes; *y*-axis: Figure 4f for right eyes, Figure 4g for left eyes), and age (*x*-axis: Figure 4d, *y*-axis: Figure 4h). The distribution plots of each variable are shown on the diagonal axis. Significant correlations were denoted as ***, **, and * for *p*-values < 0.001, <0.01, and <0.05, respectively. A significant negative correlation was observed between binocular BCVA and fixational disparities. However, no significant correlations were found between the other clinical characteristics (including age and stereoacuity) and fixational disparities (Figure 4).

## 4. Discussion

Fixational eye movements entail slow oscillations of the eyes controlled by subcortical and cortical areas of the brain [18]. Precise oculomotor control ensures the central placement of the visual target on the fovea, crucial for optimal visual acuity [19]. Fixational eye movements, which include microsaccades, drifts, and tremors, play a role in fine-tuning this alignment. Research has shown that these minuscule eye movements are generally conjugate, meaning that they move together in both eyes, and they help reduce fixational disparity by correcting for small misalignments [20,21,22,23].

Strabismus patients often experience reduced visual acuity, which is linked to impaired eye movement control [24,25]. Some researchers have found that strabismus patients may develop specific neural adaptation mechanisms over time to cope with abnormal eye movement patterns [26,27]. In severe cases of strabismus, patients may lose normal binocular vision, leading to significant abnormalities in their eye movement patterns [12,28,29].

Zhou et al. utilized the binocular eye-tracking technique to investigate eye movement functions in children with amblyopia and successfully rehabilitated amblyopia [30]. In our study, we applied the same binocular eye tracking technique to quantitatively evaluate fixational disparities in children with strabismus and healthy children, in addition to use in those with strabismus before and after corrective surgery. Disparities in sustained vision relative to the target were measured as the mean from both eyes. We found significantly increased fixational disparities in children with strabismus compared with the results for the healthy group, aligning with the results of other studies on fixational eye movements in strabismus [31,32,33].

Comparing esotropia and exotropia in the strabismus group, we found fewer fixational disparities in children with exotropia, which may be the result of there being more participants with intermittent exotropia, with them having better binocular visual function and better fixation [34]. In addition, in our study, we also evaluated fixational disparities in strabismic children following strabismus surgery to explore whether fixational eye movement deficits would be restored by surgery performed during childhood [13,35,36,37]. In the present study, we found that fixational disparities were significantly diminished in the horizontal direction following strabismus surgery. This finding robustly confirms the notion that strabismus surgery improves binocular coordination in patients and that the significant improvement in the horizontal direction might be attributed to the predominance of horizontal strabismus cases in this study [38].

The results of previous studies have indicated that strabismus can impact stereopsis and that strabismus surgery can restore binocular balance and subsequently improve stereopsis [39]. In the present study, we found that both fixational disparities and stereopsis in the strabismus group were markedly worse than those of the control group. Moreover, we also found a notable improvement in both fixational disparities and stereopsis after strabismus surgery in patients with strabismus, which was consistent with the results of a previous study. Although the stereoacuity of participants in the strabismus group was significantly improved after surgery, it was still worse than that of the healthy participants, which may be the result of specific psychological factors and other factors including suppression of one eye, limiting binocular input even after alignment, residual eye misalignment, and adaptation difficulties [40,41,42,43,44]. Additionally, we found no significant correlation between fixational disparities and stereopsis in the strabismus group, indicating the necessity of further investigations involving larger and more diverse patient groups.

Furthermore, a significant negative correlation was observed between BCVA and fixational disparities in this study. The authors of the majority of studies report a positive correlation between BCVA and fixational eye movements in patients with amblyopia [30,45]. To the best of our knowledge, this study is the first study to identify this correlation in strabismic children, and this finding suggests that strabismic children with better BCVA may exhibit fewer fixational disparities. Additionally, there was no significant interaction between age and fixational disparities, which may be due to the complex interplay among strabismus, visual development, and the variable intervals from abnormal visual experiences to the initiation of treatment.

This study evidently has some limitations that need to be addressed. Firstly, the sample size is small; however, the aim of this preliminary study is to identify trends that warrant further investigation in future clinical trials or large-scale prospective observational studies. Secondly, compared to the EyeLink(SR-Research Ltd., Mississauga, Ontario, Canada), a high-resolution eye tracker with a sampling rate of 500–2000 Hz, the Tobii Eye Track 5 with a lower sampling rate is unsuitable for tasks requiring precise temporal measurements such as saccadic kinematics or other temporal measurements; however, it is suitable for pupil measurement and spatial outcome measures such as fixational disparities [31,46]. For future studies, a direct comparison between Tobii Eye Track 5 and EyeLink during various visual tasks could offer stronger evidence.

To the best of our knowledge, this study is the first study to involve the use of an eye-tracking device to track preoperative and postoperative fixational disparities among patients with strabismus. We successfully utilized the low-cost eye-tracking device to identify abnormal fixational eye movement patterns in these patients and found that fixational functions can be significantly improved in the horizontal direction through surgical intervention.

While this study provides important insights into fixational disparities in strabismus, several avenues for future research remain. First, longitudinal studies are needed to evaluate the long-term effects of surgical alignment on fixational stability and binocular function. Such studies could help determine whether early surgical intervention leads to sustained improvements in fixational control and visual outcomes. Second, the role of vision therapy in reducing residual fixational disparities post-surgery warrants further investigation. Combining surgical alignment with targeted binocular training may optimize functional outcomes in children with strabismus.

## 5. Conclusions

In this study, we quantitatively analyzed the impact of strabismus and strabismus surgery on fixational disparities using the binocular eye-tracking technique. The results of our study showed that children with strabismus exhibited more significant fixational disparities than children in the healthy group. Moreover, fixational disparities were significantly diminished after strabismus surgery. The present results highlight the value of the binocular assessment of fixational disparities in strabismus management.

## Figures and Tables

**Figure 1 jemr-18-00006-f001:**
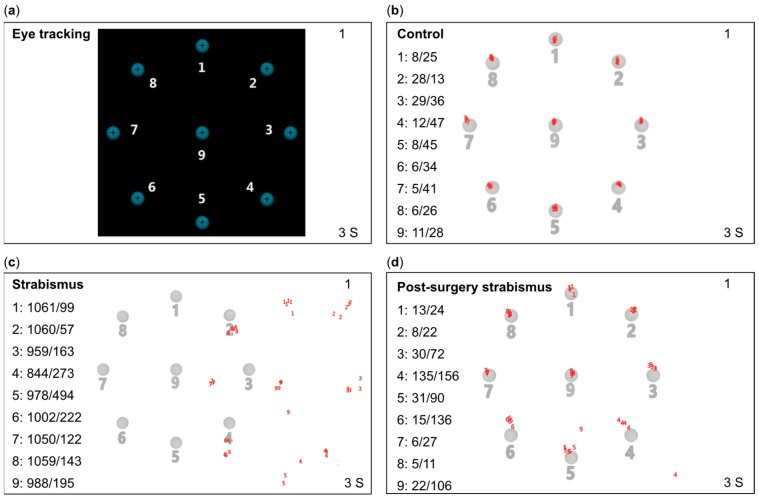
Schematic representation of the eye movement assessment paradigm and representative results from the control and strabismus groups. (**a**) Experimental paradigm: Participants fixated on a central blue target (1.4° diameter) with a black cross-shaped center, presented sequentially at nine locations: one central point and eight peripheral positions arranged on a circle (8.3° radius). Each target was displayed for 3 s. (**b**) Gaze sampling protocol: At each location, 10 consecutive gaze samples were recorded. Horizontal and vertical disparities were computed as the mean deviation (in degrees of visual angle, °) from the target position. (**c**) Preoperative results (strabismus group): Significant fixational disparities (red markers) were observed across peripheral and central targets, with mean horizontal and vertical deviations exceeding control group values (*** *p* < 0.001). (**d**) Postoperative results (strabismus group). Data and images were acquired using the Tobii Eye Tracker 5 hardware and processed with Tobii Pro Lab software (version 1.145, Tobii Technology AB, Stockholm, Sweden). The figure is published under a CC BY 1.0 license (https://creativecommons.org/publicdomain/zero/1.0/, URL accessed on 4 March 2025). No modifications were made to the original images.

**Figure 2 jemr-18-00006-f002:**
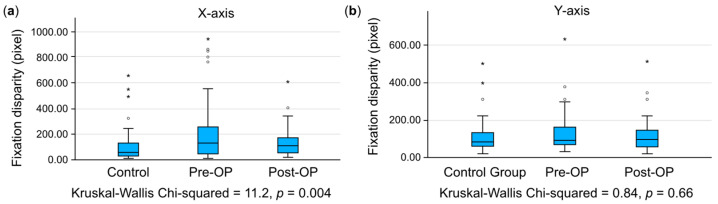
Comparison of the main outcome measures across groups. Data are presented as boxplots, where the central line represents the median, and the box edges indicate the interquartile range. Outliers are shown as individual points. Statistical comparisons among the three groups were performed using the Kruskal–Walli’s test. Significant differences in pairwise comparisons are indicated by * (*p* < 0.05), ** (*p* < 0.01) and *** (*p* < 0.001), while *p*-values > 0.05 indicate no significant differences. Error bars represent the standard error of the mean (SEM), providing an estimate of the uncertainty around the mean values.

**Figure 3 jemr-18-00006-f003:**
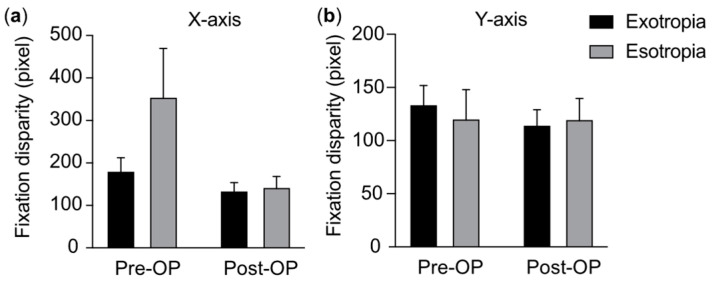
Comparison of fixational disparities between children with exotropia and children with esotropia in the strabismus group.

**Figure 4 jemr-18-00006-f004:**
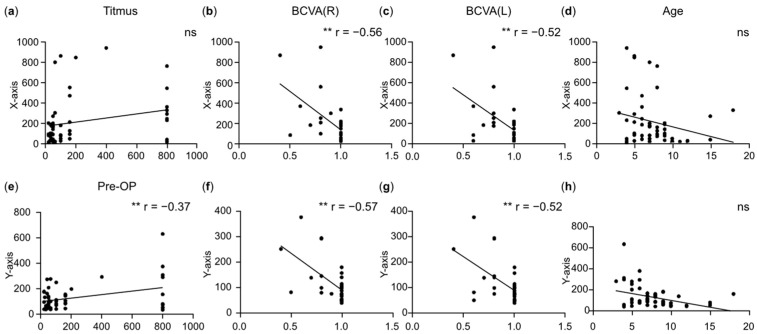
Correlations between fixational disparities and clinical characteristics in the strabismus group. (**a**–**d**) Correlations between X-axis disparity and stereoacuity (**a**), (BCVA) of the right eye (**b**), BCVA of the left eye (**c**), and age (**d**). (**e**–**h**) Correlations between Y-axis disparity and stereoacuity (**e**), BCVA of the right eye (**f**), BCVA of the left eye (**g**), and age (**h**). The distribution plots of each variable are shown on the diagonal axis. Significant correlations were marked as ** *p* < 0.01, while ns represents no significant correlation (*p* > 0.05).

**Table 1 jemr-18-00006-t001:** Demographics of participants.

Category	Count		
	Control Group	Strabismus Group	Sig.
Age range (years)	4–12	4–18	
Mean age [SD]	7.37 [0.20]	7.76 [3.59]	0.82
Sex			
Male	26 (43.3%)	36 (63.2%)	
Female	34 (56.7%)	21 (36.8%)	
Total	60	57	0.78

**Table 2 jemr-18-00006-t002:** Demographics of participants and main outcome measures of fixational disparities.

Category			Strabismus Group		Control Group		Sig.
			Mean	SD	Mean	SD	
Fixational disparities (pixel)	*X*-axis	Pre-OP	214.61	32.14	103.78	16.43	<0.001
		Post-OP	142.15	15.48			
		Sig.	0.03				
	*Y*-axis	Pre-OP	133.97	14.67	96.6	6.37	<0.001
		Post-OP	123.69	12.64			
		Sig.	0.42				
Stereoacuity (“)		Pre-OP	243.38	40.44	50.33	4.83	0.02
		Post-OP	221.29	40.19			
		Sig.	<0.001				

**Table 3 jemr-18-00006-t003:** Comparison of fixational disparities between children with exotropia and children with esotropia in the strabismus group.

Category		Pre-OP		Post-OP		Sig.
		Mean	SD	Mean	SD	
Exotropia	*X*-axis(pixel)	181.52	30.87	135.14	18.84	0.18
Esotropia		356.02	113.63	143.18	24.94	0.04
	Sig.	0.04		0.84		
Exotropia	*Y*-axis (pixel)	134.27	17.71	114.86	14.31	0.21
Esotropia		120.86	27.30	120.18	19.48	0.48
	Sig.	0.73		0.86		

## Data Availability

Data are contained within the article.

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
