# Peer review of "Quantitative Assessment of Fixational Disparity Using a Binocular Eye-Tracking Technique in Children with Strabismus"

_1995-8692, 2025, doi:10.3390/jemr18020006_

Round 1

Reviewer 1 Report

Comments and Suggestions for Authors

Methods: Refractive errors can play a role to provide optimal VA, consider to show refractive errors and relate them to fixation disparity, same for angle of strabismus

Procedure: was there a fixed instruction, was a trial inserted before measurement?   

Figure 5: more information is needed to this figure, explain what is shown in a-h, what is pre-op (titmus?) f, g pre-op? 

Discussion lines 182-185

Comments on the Quality of English Language

Quality of English can be improved (native reader?)

Author Response

Dear Reviewer,

Thank you very much for your thoughtful and constructive feedback on our manuscript. We appreciate the time and effort you have taken to review our work. Below, we have addressed each of your comments in detail and made the necessary revisions to the manuscript.

3. Point-by-point response to Comments and Suggestions for Authors

Comments 1:

Methods: Refractive errors can play a role to provide optimal VA, consider showing refractive errors and relate them to fixation disparity, same for angle of strabismus

Response 1: Thank you for pointing this out. We fully agree that refractive errors can significantly impact visual acuity (VA) and fixation disparity. Methods included objective measurements using autorefraction and subjective refinement with trial lenses. Detailed results are presented in – page 2, paragraph 4, and line 74-76.]

“[updated text in the manuscript: All participants underwent refractive correction, and measurements of fixation disparity and strabismus angles were taken both before and after correction to ensure data accuracy. ]”

Comments 2:

Procedure: was there a fixed instruction, was a trial inserted before measurement?  

Response 2: A standardized protocol was followed during measurements, with all participants instructed to fixate on a specific target and maintain a stable head position to ensure reproducibility. This step helped enhance the reliability of our measurements. Detailed results are presented in – page 3, paragraph 3, and line 115-117.]

“[updated text in the manuscript: Additionally, a trial adaptation period was included before formal measurements to familiarize participants with the testing environment and stabilize their visual state.]”

Comments 3:

Figure 5: more information is needed to this figure, explain what is shown in a-h, what is pre-op (titmus?) f, g pre-op?  

Response 3: We agree that additional information is needed to clarify Figure 5 (Now Figure 4). revised the figure caption and added a detailed explanation in the Results section – page 6, paragraph 3, and line 204-208.]

“[updated text in the manuscript: Pearson correlation analysis was performed between fixational disparities and various clinical characteristics before surgery, including stereoacuity (x-axis: Figure 4a, y-axis: Figure 4e), best-corrected visual acuity (BCVA; x-axis: Figure 4b for right eyes, Figure 4c for left eyes; y-axis: Figure 4f for right eyes, Figure 4g for left eyes), and age(x-axis: Figure 4d, y-axis: Figure 4h).]”

Comments 4:

Discussion lines 182-185

Response 3: Thank you for pointing this out. We have, accordingly, remove this description in the revised manuscript. Please refer to page 7, paragraph 2, and line 218.]

Comment 5:

Comments on the Quality of English Language Quality of English can be improved (native reader?)

Response to Comments on the Quality of English Language: Thank you for your suggestion. We have carefully reviewed the manuscript for language and clarity. Additionally, the revised manuscript has been professionally edited by MDPI’s language editing service to ensure high-quality English.

Once again, we sincerely thank you for your valuable feedback, which has significantly improved the quality of our manuscript. We hope that our revisions have addressed all your concerns, and that the manuscript is now suitable for publication.

Please do not hesitate to contact us if you have any further questions or require additional clarifications.

Best regards,

Sincerely, 

Yongchuan Liao 

Reviewer 2 Report

Comments and Suggestions for Authors

please see attachment.

Author Response

Dear Reviewer,

Thank you very much for your thoughtful and constructive feedback on our manuscript titled "Quantitative Assessment of Fixational Disparity Using a Binocular Eye-tracking Technique in Children with Strabismus" (Manuscript ID: jemr-3436674). We greatly appreciate the time and effort you have taken to review our work and provide detailed comments to help improve the quality of our paper. We have carefully addressed each of your suggestions and have revised the manuscript accordingly. Below, we provide a point-by-point response to your comments.

Comment 1:

The format needs to be adjusted to include Hou, Chen, Yang, and Liao in the upper right of each page. 

Response: We sincerely apologize for this oversight. We have now revised the manuscript to include the authors' names (Hou, Chen, Yang, and Liao) in the upper right corner of each page, as per the journal’s formatting guidelines.

Comment 2:

Please add additional information about the ethics and specify whether children signed or verbally gave assent and guardian/parents gave written consent. 

Response: Thank you for pointing this out. We have added detailed information about the ethical approval process in the revised manuscript. Specifically, we have clarified that written consent was obtained from the guardians/parents of all participating children, and verbal assent was provided by the children themselves. This information has been included in the Methods section under the Ethical Approval subsection. The additional information can be found – page 2, paragraph 3, and line 60-61.]
“[updated text in the manuscript: Written consent was obtained from the guardians/parents of all participating children, and verbal assent was provided by the children themselves.]”

Comment 3:

For Design, please add exclusion criteria. Were participants with attentional dysfunction such as attention deficit disorder (ADD) or any neurological dysfunctions (concussion, learning disabilities, etc.) excluded? If so, please include that information. 

Response: We appreciate this suggestion. We have now added a detailed description of the exclusion criteria in the Design section. Participants with attentional dysfunctions (e.g., ADD) or neurological conditions (e.g., concussion, learning disabilities) were excluded from the study. This information has been explicitly stated to provide a clearer understanding of the participant selection process. The revised exclusion criteria can be found – page 3, paragraph 2, and line 96-104.]

“[updated text in the manuscript:

Exclusion Criteria

Participants were excluded from the study if they had attentional dysfunction (e.g., attention-deficit disorder or attention-deficit/hyperactivity disorder), neurological conditions (e.g., concussion, traumatic brain injury, epilepsy, or learning disabilities), significant ocular pathologies (e.g., cataracts, glaucoma, or retinal disorders), systemic conditions (e.g., Down syndrome, cerebral palsy, or other developmental disorders), or an inability to cooperate with the eye-tracking procedure due to behavioral or cognitive limitations. These exclusion criteria were implemented to ensure that the study results were not confounded by factors unrelated to strabismus or fixational disparity.]”

Comment 4:

For Materials, you have included the temporal resolution of 120Hz. Please also include the spatial resolution of the eye tracker. What was the smallest degree of resolution, which is critical for a study on fixation disparity? 

Response: Thank you for highlighting this important detail. We have now added the spatial resolution of the eye tracker to the Materials section. The smallest degree of resolution, which is critical for a study on fixation disparity, has also been specified. The revised discussion can be found – page 3, paragraph 3, and line 109-111.]

“[updated text in the manuscript: A 32-inch 3-dimension (3D) monitor (resolution 1920×1080 pixels at a refresh frequency of 120 Hz and a spatial resolution of 0.3 degrees (minimum 0.1 degrees); LG Electronics, Seoul, Korea) was used to present stimuli at a viewing distance of 80 cm.]”

Comment 5:

You need to add a statistical analysis section to describe how your statistics were calculated. How did you correct for multiple comparisons? Were the data normally distributed, and hence parametric statistics could be used, or did you have a non-normal distribution, in which case you needed nonparametric statistical tests? 

Response: We thank the reviewer for this valuable suggestion. We have now added a dedicated Statistical Analysis section to the Methods. The revised discussion can be found – page 4, paragraph 2, and line 142-152.]

“[updated text in the manuscript: Statistical analyses were performed using SPSS (version 26.0, IBM Corp., Armonk, NY, USA). The number of shifted pixels in fixational disparities was treated as n-dimensional data and expressed as the mean [standard deviation]. To address potential issues of multiple comparisons, appropriate corrections (e.g., Bonferroni or false discovery rate adjustments) were applied where necessary. The normality of the data distribution was assessed using statistical tests and visual inspections. For data that followed a normal distribution, parametric tests were used; for data that did not follow a normal distribution, nonparametric tests were employed. Specifically, the Mann–Whitney U test was used for comparisons between groups, and paired sample tests (e.g., paired t-tests or Wilcoxon signed-rank tests, depending on the distribution) were used to compare internal differences within the same group. Differences were considered statistically significant at P < 0.05.]”

Comment 6:

On line 109, I think when you use SD, you mean standard deviation, whereas the paper states standard disparity. 

Response: We sincerely apologize for this error. We have corrected the term standard disparity to standard deviation (SD) on line 109 to ensure consistency with standard statistical terminology. The revised information can be found – page 4, paragraph 2, and line 144.]

“[updated text in the manuscript: The number of shifted pixels in fixational disparities was treated as n-dimensional data and expressed as the mean [standard deviation].] “

Comment 7:

For Figure 3, please add the units for fixation disparity on the y-axis. Please describe what the error bars mean (standard deviation, standard error of the mean, or 95% confidence range). 

Response: Thank you for pointing this out. We have revised Figure 3 (now for Figure 2) to include the units for fixation disparity on the y-axis. Additionally, we have clarified in the figure caption that the error bars represent the standard error of the mean (SEM). Please refer to Figure 2 in the revised manuscript. The revised figure caption can be found – page 5, paragraph 2, and line 178-179.]

“[updated text in the manuscript: The error bars in Figure 2 represent the standard error of the mean (SEM), which pro-vides an estimate of the uncertainty around the mean values.] “

Comment 8:

For the Discussion, please include a section on future directions. 

Response: We appreciate this suggestion. We have added a new subsection to the Discussion titled Future Directions. The revised discussion can be found – page 9, paragraph 2, and line 295-303.

“[updated text in the manuscript: While this study provides important insights into fixational disparities in strabismus, several avenues for future research remain. First, longitudinal studies are needed to evaluate the long-term effects of surgical alignment on fixational stability and binocular function. Such studies could help determine whether early surgical intervention leads to sustained improvements in fixational control and visual outcomes. Second, the role of vision therapy in reducing residual fixational disparities post-surgery warrants further investigation. Combining surgical alignment with targeted binocular training may optimize functional outcomes in children with strabismus.] “

Conclusion

Once again, we would like to express our sincere gratitude for your insightful comments and suggestions, which have significantly improved the quality of our manuscript. We hope that the revisions address all your concerns, and that the manuscript is now suitable for publication in the Journal of Eye Movement Research. Please do not hesitate to contact us if you have any further questions or require additional clarifications.

Thank you for your time and consideration.

Sincerely, 

Yongchuan Liao 

Reviewer 3 Report

Comments and Suggestions for Authors

Author Response

Dear Reviewer,

Thank you very much for your thoughtful and constructive feedback on our manuscript titled "Quantitative Assessment of Fixational Disparity Using a Binocular Eye-tracking Technique in Children with Strabismus" (Manuscript ID: jemr-3436674). We greatly appreciate the time and effort you have taken to review our work and provide detailed comments to help improve the quality of our paper. We have carefully addressed each of your suggestions and have revised the manuscript accordingly. Below, we provide a point-by-point response to your comments.

Positive Comments:

- Large amount of participants

- Solid statistical evaluation

Response 1: Thank you for recognizing the strength of our sample size. We also appreciate your acknowledgment of our statistical methods.

Negative Comment:

Stereoacuity (Table 2) seems to be bad even after the surgery, and there is no information explaining why this is the case. 

Response: We sincerely thank the reviewer for raising this important point. We have added a detailed discussion of these potential reasons in the revised manuscript to provide a clearer explanation for the observed results. The revised discussion can be found – page 8, paragraph 2, and lines 259 to 264.]

“[updated text in the manuscript: Although the stereoacuity of participants in the strabismus group was signifi-cantly improved after surgery, it was still worse than that of the healthy par-ticipants, which may be the result of specific psychological factors and other factors including suppression of one eye, limiting binocular input even after alignment, residual eye misalignment, and adaptation difficulties [40-44]”

Minor Comment 1:

Results could have been reported with whisker plots. 

Response: We appreciate this suggestion. In the revised manuscript, we have replaced the current graphical representations with whisker plots (e.g., box-and-whisker plots) for better visualization of the data distribution, including median, interquartile range, and outliers. This change enhances the clarity and interpretability of the results. The revised Figure 2 can be found – page 5, paragraph 4, and line 170.]

Minor Comment 2:

For the control group, Pre-OP and Post-OP does not apply; it would be better to use only one value here. 

Response: We agree with the reviewer’s observation. In the revised manuscript, we have modified the presentation of the control group data to include only one value (e.g., a single baseline measurement) instead of splitting it into Pre-OP and Post-OP. This adjustment ensures consistency and avoids unnecessary confusion. The revised Table can be found – page 5, paragraph 2, and line 169.]

Overall Comment:

For me, this paper is a solid study and evaluation of Pre-OP and Post-OP. The only thing which is missing is the explanation why the stereoacuity is still bad after the OP. Overall, I would argue for accepting this paper after a small revision. 

Response: We sincerely thank the reviewer for their positive feedback and constructive suggestions. We have addressed all the comments, including the explanation for the poor stereoacuity after surgery, the use of whisker plots, and the adjustment of control group data presentation. We believe these revisions have significantly improved the manuscript, and we hope it now meets the journal’s standards for publication. 

We are grateful for the opportunity to improve our manuscript and hope that the revised version addresses all the reviewer’s concerns. Thank you again for your valuable feedback.

Thank you for your time and consideration.

Sincerely, 

Yongchuan Liao 
